# Fundamental Studies on Crystallization and Reaching the Equilibrium Shape in Basic Ammonothermal Method: Growth on a Native Lenticular Seed

**DOI:** 10.3390/ma15134621

**Published:** 2022-06-30

**Authors:** Tomasz Sochacki, Robert Kucharski, Karolina Grabianska, Jan L. Weyher, Malgorzata Iwinska, Michal Bockowski, Lutz Kirste

**Affiliations:** 1Institute of High Pressure Physics of thePolish Academy of Sciences, Sokolowska 29/37, 01-142 Warsaw, Poland; kucharski@ammono.pl (R.K.); kgrabianska@unipress.waw.pl (K.G.); weyher@unipress.waw.pl (J.L.W.); miwinska@unipress.waw.pl (M.I.); bocian@unipress.waw.pl (M.B.); 2Fraunhofer Institute for Applied Solid State Physics (IAF), Tullastraße 72, 79108 Freiburg, Germany; lutz.kirste@iaf.fraunhofer.de

**Keywords:** GaN, crystal growth, basic ammonothermal method, X-ray topography, high-resolution X-ray diffraction, diffuse scattering

## Abstract

In this paper, a detailed investigation of the basic ammonothermal growth process of GaN is presented. By analyzing the crystallization on a native seed with a lenticular shape, thus with an intentionally varying off-cut, we wanted to answer some basic questions: (i) Which crystallographic planes play the most important role during growth (which planes are formed and which disappear)? (ii) What is the relationship between the growth rates in different crystallographic directions? (iii) What is the influence of the off-cut of the seed on the growth process? Two non-polar slices, namely, $\left(1\bar{2}10\right)$ and $\left(\bar{1}100\right)$, as well as a $\left(0001\right)$ basal plane slice of an ammonothermal crystal were analyzed. The examined planes were selectively etched in order to reveal the characteristic features of the growth process. The applied characterization methods included: optical microscopy with Nomarski contrast and ultraviolet illumination, X-ray topography and high-resolution X-ray diffraction, and secondary ion mass spectrometry. The obtained results allowed for creating a growth model of an ammonothermal GaN crystal on a lenticular seed. These findings are of great importance for the general understanding of the basic ammonothermal crystal growth process of GaN.

## 1. Introduction

Gallium nitride (GaN) substrates of high structural perfection are needed for advanced electronic and optoelectronic devices based on GaN-on-GaN technology [1,2]. The wafers can be prepared from crystals grown using three main methods: crystallization from the gas phase at ambient pressure [3,4,5], basic or acidic high-pressure ammonothermal growth [6,7], and growth from a solution of gallium and sodium under nitrogen pressure [8]. One of the most promising GaN crystallization methods is the basic ammonothermal method [9]. It allows for crystalizing highly conductive (n-type) and semi-insulating ammonothermal GaN (Am-GaN). Recently, 2-inch Am-GaN wafers of extraordinary structural quality were introduced to the market [10]. They are crystallographically flat, thus with a uniform off-cut and the threading dislocation density (TDD) does not exceed 5 × 10^4^ cm^−2^ [11]. Despite many published papers devoted to crystallization [12], as well as to the properties of ammonothermal crystals [13], a detailed investigation of the basic ammonothermal growth process has never been presented. With this paper, we aim to fill this gap. By analyzing crystallization on a native seed of a lenticular shape (lens seed), thus with a varying off-cut on its surface, we aimed to answer some basic questions: (i) In which crystallographic directions does the growth proceed and which crystallographic planes play the most important role (which are formed and which disappear in time)? (ii) What are the relationships between growth rates in different crystallographic directions? (iii) What is the influence of the off-cut of the seed on the growth process? For this purpose, slices of two crystallographic non-polar planes, namely, $\left(1\bar{2}10\right)$ and $\left(\bar{1}100\right)$, of a crystal grown on a lenticular seed (also called a lens seed) were prepared and analyzed. Photo-etching (PE) under UV light, optical microscopy (OM) with Nomarski contrast and ultraviolet (UV) illumination, and X-ray topography (XRT) and high-resolution X-ray diffraction (HRXRD) were applied as the main methods for investigating structural properties, such as growth striations [14] and dislocations. Furthermore, secondary ion mass spectrometry (SIMS) was used for analyzing concentrations of impurities in all sectors of the analyzed crystal slices with different crystallographic orientations. In addition, from a part of the newly grown crystal, a $\left(0001\right)$ slice was cut that was subjected to defect selective etching (DSE) in molten eutectic KOH-NaOH in order to analyze the etch pit density (EPD). The obtained results allowed us to create a growth model of Am-GaN crystallized on a lens seed, which is of general importance for the ammonothermal growth process of GaN.

## 2. Methods

An n-type Am-GaN crystal of the highest structural quality and a diameter of 26 mm was selected as a native seed for a standard basic ammonothermal growth run. This crystal was grown in two subsequent ammonothermal processes. Its $\left(000\bar{1}\right)$ surface was prepared to provide a lenticular shape (see Figure 1) via lapping and mechanical and chemo-mechanical polishing (CMP). The opposite $\left(0001\right)$ surface was optically flat. The thickness of the seed in its center was 4.1 mm and the radii of the surface curvature varied between 21 and 25 mm.

The seed was attached by its $\left(0001\right)$ surface to a special metal holder and placed with other standard native seeds (not of lenticular shape) into an autoclave for ammonothermal GaN growth. Then, an ammonothermal crystallization run was performed with a twice higher growth rate than in a typical process. The details of a standard basic ammonothermal growth process, as well as the crystal growth zone configuration, including attaching the seeds to metal holders, were described elsewhere [9,11,12,15]. After the crystallization run (lasting 64 days), the selected seed with the newly grown crystal on it was examined. The obtained crystal was sliced in such a way that sample slices with the following crystallographic planes were obtained: $\left(1\bar{2}10\right)$, $\left(\bar{1}100\right)$ and $\left(0001\right)$. These slices were subjected to lapping, mechanical polishing and CMP to an epi-ready state. Non-polar samples were analyzed using OM with UV light and then treated via photo-eching (PE) in a modified KSO-D solution (0.02 M K_2_S_2_O_8_ + 0.02 M KOH) [16,17] with the addition of a 0.02 M Na_3_PO_4_ component in order to increase the stability of the solution [18]. A galvanic mode (a GaN sample connected to a Pt electrode via a Ti spring) was employed for revealing electrically active inhomogeneities, such as striations. PE was performed under the illumination of a 300 W UV-enhanced Xe lamp (Oriel, Germany) [19,20]. DSE was performed on the (0001) surface in molten eutectic KOH-NaOH with 10% of MgO at 500 °C (temperature of the hot plate) for 10–20 min [21]. The dislocation density was established by counting the overall number of etch pits (EPD). The analysis was performed only in terms of the pit density and not their size. Therefore, no information on the density of different types of dislocations, namely, screw, mixed and edge, which are correlated with the pit size, was gathered [22]. As mentioned, all surfaces (epi-ready, after DSE and following PE) were characterized with a Nikon Eclipse LV100ND OM with Nomarski contrast and under UV illumination.

For the XRT analysis of the crystals’ defect structure, a RIGAKU XRTmicron laboratory camera was used (Rigaku, Tokyo, Japan). The camera was equipped with a high-brilliance microfocus X-ray source combined with multilayer X-ray optics. For the imaging, Cu-Kα_1_ radiation (λ = 154.06 pm, 8.05 keV) and Mo-Kα_1_ radiation (λ = 70.94 pm, 17.48 keV) were applied. The XRT analysis was performed in a transmission geometry (Lang technique) via exposures using 0002 type reflections for both $\left(1\bar{2}10\right)$ and $\left(\bar{1}100\right)$ samples, as well as $\bar{3}030$ and $11\bar{2}0$ reflections for $\left(1\bar{2}10\right)$ and $\left(\bar{1}100\right)$ slices, respectively. The X-ray topographs were recorded with a high-resolution CCD camera (5.4 μm pixel size). Values of µt (µ is the linear absorption coefficient and t is the crystal thickness) were 8.6–10.3 and 8.0–9.6 for the Cu-Kα_1_ and Mo-Kα_1_ radiation, respectively. This meant that the XRT measurements were performed under Borrmann contrast conditions, as described in [13] and the literature referred to therein. HRXRD was applied to analyze the lattice parameters and mosaicity in various crystal regions. A Panalytical MRD system equipped with a 4 × Ge 220 Bartels monochromator (Cu-Kα_1_-radiation) and a 3 × Ge 220 analyzer was used (Panalytical, Almelo, The Netherlands). For each crystal slice, symmetric and asymmetric reflections were used at six different positions to perform 2Θ/Θ-, 2Θ/ω- and ω-scans, as well as reciprocal space maps (RSMs). In order to limit the measured spot to a specific sample location, a pinhole aperture with a diameter of 1.5 mm was used. For the calculation of the lattice parameters, refraction-corrected data were applied [23]. SIMS measurements were performed in selected areas of the $\left(1\bar{2}10\right)$ and $\left(\bar{1}100\right)$ surfaces to determine the contamination of Am-GaN in the different oriented slices, as well as in different growth sectors. A CAMECA IMS6F microanalyzer was used. Molecular oxygen and cesium ions were applied as primary ions. The generated secondary ions were detected with a mass spectrometer, and the relative sensitivity factors derived from standard samples were used for quantitative calibration of the secondary ion intensities.

## 3. Results

Figure 2a presents a crystal with a hexagonal shape (plane view) that was grown on the lens seed (see Figure 1). The clearly visible side walls were sloped toward the $\langle \bar{1}100\rangle$ direction. The average total thickness of the whole crystal (seed and the new-grown material) was 6.92 mm. The average growth rate in the $\left[000\bar{1}\right]$ direction was 44 µm/day. The maximum crystal widths in the $\langle \bar{1}100\rangle$ and $\langle 1\bar{2}10\rangle$ directions (the lateral size of the crystal) were 26 and 29 mm, respectively. The white dashed lines presented in Figure 2a indicate the slicing locations that were used to obtain samples with the non-polar $\left(1\bar{2}10\right)$ and $\left(\bar{1}100\right)$ planes. The white solid trapezoid represents an additional sample obtained by cutting. The slicing was performed in the $\left(000\bar{1}\right)$ plane, just below the top of the as-grown surface of the hexagonal crystal. The $\left(0001\right)$ surface of this sample, shown in Figure 2b, was prepared to an epi-ready state and subjected to DSE. The EPD was determined at the five points marked as a1, a2, c, m1, and m2 in Figure 2b.

Figure 3 shows slices under UV illumination of the two analyzed non-polar planes $\left(1\bar{2}10\right)$ and $\left(\bar{1}100\right)$. The intensity and color of the luminescence allowed for distinguishing the five growth areas divided and marked with white dashed lines. These were named with Arabic numerals in brackets for description and discussion. Area (0) shows the first Am-GaN seed, called the “pre-seed”. Area (1) represented the proper Am-GaN seed prepared to provide a lenticular shape. Blue luminescence at the interface between the pre-seed and the proper seed (lens seed) was clearly visible. Areas (2)–(4) represented three different parts of the newly grown crystal. Area (2) was where the first stage of growth was realized and the $\left(000\bar{1}\right)$ plane was recovered. Herein, we could distinguish two sub-areas: (2a) and (2b). Sub-area (2a) was dominated by strong blue luminescence, whereas sub-area (2b) did not show this feature. Area (3) showed the brightest green luminescence. It represented the part of the crystal that grew in the lateral directions. Area (4) marked the crystal growth in the $\left[000\bar{1}\right]$ crystallographic direction. It should be noted that the two presented slices had completely different shapes. Similarities were only found in the intensity and color of the luminescence of the five areas and the interfaces between them. Another common feature was a crack for both slices in area (4), starting from the boundary of area (3). Roman numerals were used to designate the six locations of HRXRD and SIMS measurements called “analysis points (APs)”.

Figure 4a shows the $\left(1\bar{2}10\right)$ slice after the PE. It is possible to distinguish four areas with varying degrees of etching. Areas (0), (1) and (4) were photo-etched at the same rate. Sub-area (2a) was etched at a higher rate than the areas mentioned above, as well as sub-area (2b). Parts of area (3) were not photo-etched or were etched very slowly. Four sectors of the slice sample were chosen for further analysis. They are marked as rectangles in Figure 4a and named: A, B, C and D. These sectors were chosen in order to include interfaces between two or even three selected areas. Figure 4b–e represent the magnifications of the four selected regions: A–D. In sector A (see Figure 4b), some striations parallel to the $\left(000\bar{1}\right)$ plane were clearly visible in area (1). Similar striations were also noted in area (2a). They were placed parallel to the interface (marked with a dashed line) between areas (1) and (2). In area (2b), the striations created an inclination angle of 43.1°. to the $\left(000\bar{1}\right)$ plane that corresponded to the $\left(\bar{1}01\bar{2}\right)$ plane. This change in the slope was accompanied by a change from area (2) to an area of lateral growth (3). Figure 4c shows a part of the slice from sector B. One can observe striations that were parallel to the interface with area (1) and which enter into area (4) at a certain angle. It should be noted that the interface between areas (2) and (4) was not parallel to the $\left(000\bar{1}\right)$ plane. Its inclination angle was about 1° Figure 4d shows a part of a slice from sector C. In area (3), a change from 43.1° to 61.9° in the inclination angle of the striations to the $\left(000\bar{1}\right)$ plane was noticed. The value of 61.9° corresponded to the $\left(\bar{1}01\bar{1}\right)$ plane. Furthermore, the edge of the whole crystal was inclined at an angle of 90° to the $\left(000\bar{1}\right)$ plane forming a $\left(\bar{1}010\right)$ facet. Figure 4e shows a part of the slice from sector D. All the visible striations had the same angle of inclination (43.1°) to the $\left(000\bar{1}\right)$ plane. A change in the interface slope between areas (3) and (4) was observed.

Figure 5a shows the $\left(\bar{1}100\right)$ slice after the PE. The lowest etching rate was observed for area (3). Some striations can be seen in the selected areas. Magnifications of the selected sectors marked A–D are presented in Figure 5b–e. In area (1), some striations parallel to the $\left(000\bar{1}\right)$ plane were visible (see Figure 5b). In area (2), above the interface (white dashed line) with area (1), the striations were placed parallel to this interface. Then, their shape changed abruptly from a curved to a straight line with an inclination angle of 39.1° to the $\left(000\bar{1}\right)$ plane that corresponded to the $\left(\bar{1}\bar{1}2\bar{4}\right)$ plane. Figure 5c shows sector B of the slice. Herein, parallel striations reached the interface with area (4) at a certain angle. The interface between areas (2) and (4) was not parallel to the $\left(000\bar{1}\right)$ plane. Its inclination angle was about 1° Figure 5d represents sector C of the slice. In area (3), a change in the inclination angle of the striations to the $\left(000\bar{1}\right)$ plane was noticed. Close to the interface, the angle of inclination was 39.1° Further away from the interface, the angle increased to 58.4°, which corresponded to the $\left(\bar{1}\bar{1}2\bar{2}\right)$ plane. Figure 5e shows sector D from the slice. In area (3), no significant change in the inclination angle of the striations was observed. All the visible striations had the same angle of inclination to the $\left(000\bar{1}\right)$ plane: 47.3°, which corresponded to the $\left(\bar{1}\bar{1}2\bar{3}\right)$ plane.

Figure 6a and Figure 7a show the $000\bar{2}$ reflection overview X-ray topographs of the $\left(1\bar{2}10\right)$ and $\left(\bar{1}100\right)$ slices, respectively. The observed structural characteristics for the two slices of different crystallographic orientations were comparable and, therefore, are discussed together. At first glance, five different areas from (0) to (4), comparable to the OM images generated under UV illumination and presented in Figure 3, were distinguished. Striations were also clearly visible in each area. They corresponded to the same striations that were already revealed by the PE (see Figure 4). In the XRT image, the bright–dark contrasts of striations were the result of a slight bending of the lattice planes due to small differences in the lattice parameters caused by the incorporations of contaminations or dopants [24]. Magnified images of sectors A, B, C and D indicated by dashed lines in Figure 6a and Figure 7a are presented in Figure 6b,c and Figure 7b,c, respectively. In both slices presented in Figure 6a and Figure 7a, white lines were visible leading from the seed area (1) to the following areas: (2), (3) and (4). These white lines corresponded to Borrmann contrasts of threading dislocations. According to the visibility criteria for dislocations and the diffraction vector used, these were threading dislocations with a screw component [25].

For both slices, close to the center of the crystal, some dislocations from area (1) went almost straight to area (4). This is clearly visible in the magnifications of sectors A and C presented in Figure 6b and Figure 7b. Moving away from the center of the crystal, the double refraction of a dislocation was observed. The first refraction appeared at the interface between areas (1) and (2a). The second one took place at the interface between areas (2a) and (4) (see Figure 6b and Figure 7b). The refraction of growth dislocations typically occurs when the dislocations penetrate a growth sector boundary. Sectors B and D (see Figure 6c and Figure 7c) were located much closer to the edge of the crystal. Here, double dislocation refractions were observed too. The first refraction occurred at the interface between area (1) and sub-area (2a). The second one took place between sub-areas (2a) and (2b). Moving toward the edge of the crystal, only one refraction was observed. It was at the interface between areas (1) and (3). On the right side of Figure 7b, the second refraction occurred between sub-areas (2a) and (2b). Figure 7c shows a magnification of sector D, which was located much closer to the edge of the crystal. Here, a two-stage dislocation refraction was observed. The first refraction occurred at the interface between area (1) and sub-area (2b). The second one took place between sub-area (2b) and area (3). A further observation was that there were striking and rather bright contrasts in the two topographs for the areas of lateral growth at the edge of the slices labeled (3). The contrasts there were not only bright but also diffuse, although light grey contrasts of the striations were still visible. One explanation would be that these areas had a high defect density. Accordingly, the Borrmann effect would be blocked, and for this reason, the X-ray topographic contrasts would appear bright. However, the results of the PE analysis did not give any indication of a high defect density in the laterally grown crystal regions, and therefore, contradict this assumption. Bright contrasts were also observed in the cracked areas. This can be explained as an orientation contrast, i.e., the corresponding crystal region was not in a diffraction condition due to the misorientation of the cracked region.

The apparently low defect density observed from the $000\bar{2}$ reflection topographs for the two slices, namely, the $\left(1\bar{2}10\right)$ and $\left(\bar{1}100\right)$ planes, did not display the entire dislocation content present since this reflection only detected threading dislocations with a screw component (pure screw or mixed-type dislocations). For this reason, additional XRT measurements were performed. To image the $\left(1\bar{2}10\right)$ and $\left(\bar{1}100\right)$ slices, the $\bar{3}030$ and $11\bar{2}0\;$reflections were used, respectively. The selected reflections were sensitive to threading dislocations with an edge component (pure edge or mixed-type dislocations). For both crystal slices, a TDD exceeding 10^4^ cm^−2^ was found in the respective topographs. This indicated that the predominant threading dislocation types in the lens-seed-grown Am-GaN crystal were edge and/or mixed-type dislocations that ran along the $\langle 0001\rangle$ direction. Furthermore, similar striations were observed in the $\bar{3}030$ and $11\bar{2}0\;$reflections, as in the type $000\bar{2}$ reflection topographs. Likewise, in the laterally grown region (3), mainly bright diffuse contrasts were also observed for these reflections. For completeness, the $\bar{3}030$ reflection topograph of the $\left(1\bar{2}10\right)$ slice and the $11\bar{2}0\;$reflection topograph of the $\left(\bar{1}100\right)$ slice are shown in Appendix A.

The HRXRD analysis showed occasionally striking differences with respect to the shape and intensity of the GaN Bragg peaks for both investigated non-polar slices, depending on the sample location. As an example, RSMs were selected from three sample locations (APs) of the *a*-plane and *m*-plane slices to describe these differences. Figure 8 shows the RSMs of the symmetrical $1\bar{2}10$ and the asymmetrical $1\bar{2}12$ reflections of the $\left(1\bar{2}12\right)$ slice and Figure 9 shows the RSMs of the symmetrical $\bar{3}300$ and the asymmetrical $\bar{2}201$ reflections of the $\left(\bar{1}100\right)$ slice. The RSMs of APs II, V and VI were selected for both samples and the observations were almost comparable despite the different crystallographic orientations of the two slices.

For this reason, the results of the RSMs of the *a*-plane and *m*-plane samples are described together. The AP IIs were located in region (4) of the newly grown Am-GaN in the $\left[000\bar{1}\right]$ direction, the AP Vs were from region (2) of the crystal where the first growth phase occurred and the $\left(000\bar{1}\right)$ plane was recovered, and the AP VIs belonged to the laterally grown region (3) (for the orientation, see Figure 3). High crystalline perfection was observed for the AP IIs of both the $\left(1\bar{2}10\right)$ and $\left(\bar{1}100\right)$ slices. The broadening of the corresponding reflections in the q_//_ reciprocal space direction was small and confirmed the high structural quality of these crystal regions. This was particularly the case for the $1\bar{2}10$ and $1\bar{2}12$ reflections of the *a*-plane sample. Furthermore, the high structural perfection in these areas, as well as the good surface quality, were evidenced by the presence of crystal truncation rods (CTRs). CTRs are dynamic X-ray scattering effects that emerge as continuous-intensity rods connecting Bragg peaks along the surface normal. In reciprocal space, CTRs are visible as broadened streaks in the q_⊥_-direction and become apparent due to the loss of translational invariance and crystal lattice order in the near-surface region [26,27]. A CTR was particularly prominent in the$\;1\bar{2}10$ reflection RSM measured at AP II of the *a*-plane slice. The slight tilt of the CTR in this measurement was due to an unintentional preparation-related miscut by ~6° of the $\left(1\bar{2}10\right)$ surface. In the RSMs of the AP Vs, the CTRs were no longer as clearly visible as in the AP IIs. Different phenomena, which overlapped, were observed in these RSMs: broadening of the reflections in the q_//_-direction connected with mosaicity, splitting of the reflections and diffuse scattering. Patterns with significantly different appearances were observed for the GaN reflections in the RSMs of the AP Vis (area (3) with lateral growth). The reciprocal lattice points were enormously broadened by the diffusely scattered intensity. The central areas of the reflections were split. Another observation for the HRXRD measurements in the AP Vis was that the reflection positions were shifted to smaller values in reciprocal space compared with the reflection positions of the other Aps. This indicated an increase in the *a*-lattice parameters for these GaN crystal regions. The RSMs of Aps I and III, which are not shown, had almost the same qualitative appearance as the RSMs of the AP Iis and demonstrated the high crystalline perfection of these GaN regions. For the RSMs of the AP Ivs, which are also not shown, reflection broadening and diffuse scattering were observed, similar to the RSMs of the AP Vs. In order to determine the *a*- and *c*-lattice parameters of the different crystal regions on the $\left(1\bar{2}10\right)$ and $\left(\bar{1}100\right)$ GaN slice, HRXRD 2Θ/Θ-scans and 2Θ/ω-scans were performed from the same symmetrical and asymmetrical reflections as used for the RSMs. The measurement profiles are shown in Figure A3a,b and Figure A4a,b of Appendix B. The determined lattice parameters are reported in Table A1 and Table A2 of Appendix C. HRXRD ω-scans were also performed to allow for a quantitative analysis of the above-discussed mosaicity and diffuse scattering. The full width at half maximum (FWHM) was determined for all six investigated APs of the two crystal slices. In addition, in order to determine a kind of quantification for the weak-intensity diffuse scattering, the full width of the thousandth maximum (FWTM) was determined in each case. The ω-scans are shown in Figure A3c,d and Figure A4c,d (Appendix B). The determined values of FWHM and FWTM are listed in Table A1 and Table A2 (Appendix C).

Table 1 and Table 2 show the results of the SIMS measurements performed on the $\left(1\bar{2}10\right)$ and $\left(\bar{1}100\right)$ slices, respectively, at six APs (see Figure 3a,b). For both samples, radical changes in hydrogen (H), sodium (Na) and oxygen (O) concentrations were observed in area (3). The concentrations of other elements only slightly fluctuated. The value for zinc (Zn) was close to 1–2 × 10^17^ cm^−3^ in all the analyzed areas. Metals such as Mg, Mn, Fe and Zn that were observed in the basic ammonothermal GaN came from ceramic and metal elements (i.e., holders) of the autoclave in the crystal growth zone. The SIMS measurements were also performed in sub-areas (2a) and (2b). The highest concentrations of H, Na and O were noted for area (2b). Furthermore, the lowest H and O concentrations were measured for area (2a).

Figure 10 shows the etch pit distribution after the DSE on the $\left(0001\right)$ plane slice in the five areas marked as: c, a1, a2, m1 and m2 in Figure 2b. It was found that the EPD in the center of the crystal and near the top of the lens seed was 3.5 × 10^4^ cm^−2^ (see Figure 10a). Halfway between the center and the edge of the crystal in m1 and a1, the EPD decreased to 7.6×10^3^ cm^−2^ and 1.6×10^4^ cm^−2^, respectively (see Figure 10b,c). At the edge of the sample in m2 and a2, the EPD decreased even more and was equal to 2.8 × 10^3^ cm^−^^2^ and 4.4 × 10^3^ cm^−2^, respectively (see Figure 10d,e).

## 4. Discussion

In the growth process, a hexagonal Am-GaN crystal of a uniform thickness was obtained from a lenticular round seed. It was thus concluded that for the applied growth conditions and the duration of the process, the crystal reached its equilibrium shape. Preliminary conclusions about how the crystal was grown were drawn from the images of the $\left(1\bar{2}10\right)$ and $\left(\bar{1}100\right)$ slices under UV illumination (see Figure 3a,b). Interfaces between the five distinguished areas were clearly visible. It was obvious that the different luminescences under UV shown in Figure 3 were determined by the different contents of impurities. The amount of non-intentionally incorporated elements depended mainly on the growth direction. The effect of the increase in the growth rate was visible in the observed lower [O] measured in area (4) in relation to the measurements performed in area (1). This resulted from the fact that the seed was crystallized with a growth rate that was half that of the growth rate of area (4). It was concluded that the level of impurities also depended on the growth rate. However, it was impossible to determine which impurity affected the luminescence under UV only on the basis of these data. The measurements of the dopant concentrations should have been supplemented with additional optical measurements. Based on the differences in the luminescence, it was concluded that each of the three newly created areas grew in a different way. The intensity and the color of the luminescences of areas (0), (1) and (4) were similar. Growth in these areas was carried out in the same $\left[000\bar{1}\right]$ crystallographic direction. The distinguishing part of the samples with the most intensive luminescence under the UV illumination was area (3). It was characterized by very high concentrations of H and Na. This part of the crystal was grown in semi-polar directions until the formation of the $\left(\bar{1}010\right)$ non-polar plane (see Figure 4 and Figure 5).

More about the growth direction in the specific areas can be discovered based on the analysis of the photo-etched cross-sections. Photo-etching is a method that is very sensitive to changes in the concentration of carriers in the material. Areas with a lower free carrier concentration are etched with a higher rate under UV light illumination than those with a higher free carrier concentration [22]. Moreover, PE allows for revealing the striations, which are related to small fluctuations in the carrier concentration at the crystallization front [23]. Thus, it can be stated that all the striations revealed using PE were arranged perpendicularly to the growth direction in the individual areas.

It should be remarked that striations are commonly formed in crystals grown from melt and solutions containing impurities. Deviations from stoichiometry or constituents of the solvent can contribute to these imperfections. Striations as micro-inhomogeneities are associated with fluctuations in growth conditions (e.g., heat and mass transfer, back-etching or regrowth, and other growth discontinuities) and thus are arranged as striations normal to the growth direction. The striations may occur in a dense sequence [14].

On both the $\left(\bar{1}100\right)$ and $\left(1\bar{2}10\right)$ slices, area (2a) was etched with the highest rate (see Figure 4c and Figure 5c). In this area, the oxygen donor concentration was the lowest (see Table 1 and Table 2). The striations visible in this area were placed parallel to the curvature of the interface with area (1). This demonstrated that in this area, the growth was carried out perpendicularly to the seed curvature. Closer to the center of the seed, it was found that the striations reached the interface with region (4) at a certain angle. On the other hand, in area (4), striations were found parallel to the $\left(000\bar{1}\right)$ plane. On the basis of the striations’ arrangements, it was concluded that in areas (4) and (1), the growth was carried out perpendicularly to the $\left(000\bar{1}\right)$ plane. It seems that in the center of the seed, where the off-cut was relatively small, the stable $\left(000\bar{1}\right)$ plane was formed in the initial stage of the ammonothermal growth process. After that, the growth was carried out in the $\left[000\bar{1}\right]$ direction. Thus, the growth in the $\left[000\bar{1}\right]$ direction first started in the center of the crystal. Area (4) then enlarged laterally as area (2a) grew.

The images after photo-etching of both slices from regions located near the edge of the seed looked similar (see Figure 4b and Figure 5b). The change in the Borrmann contrasts of the striations characteristic of area (2a) into straight lines with a specific angle of inclination relative to the plane $\left(000\bar{1}\right)$ was clearly visible.

In the analyzed $\left(1\bar{2}10\right)$ slice, the striations were inclined at an angle of 43.1°. This value corresponded to the angle between the $\left(\bar{1}01\bar{2}\right)$ and $\left(000\bar{1}\right)$ planes. In the case of the $\left(\bar{1}010\right)$ plane, the striations were inclined at an angle of 39.1°. This value corresponded in turn to the angle of inclination of the $\left(\bar{1}\bar{1}2\bar{4}\right)$ plane to the $\left(000\bar{1}\right)$ plane. The appearance of both semi-polar planes corresponded to the place on the curvature where the off-cut of the substrate’s surface was about 25° In both cases, after the formation of two semi-polar planes, the crystal started to grow perpendicularly to them.

The observed semi-polar planes were not low-energy planes. Therefore, transitions between different crystallization planes could be observed in area (3). In Figure 4d, a transition between the $\left(\bar{1}01\bar{2}\right)$ and $\left(\bar{1}01\bar{1}\right)$ planes occurred. The $\left(\bar{1}01\bar{1}\right)$ plane was tilted at an angle of 61.9° to the $\left(000\bar{1}\right)$ plane. At the periphery of the crystal, a facet that inclined at 90° to the $\left(000\bar{1}\right)$ plane was visible. This facet corresponded to the stable, low-energy, non-polar $\left(\bar{1}010\right)$ plane.

On the $\left(\bar{1}100\right)$ slice, the $\left(\bar{1}\bar{1}2\bar{4}\right)$ plane could be found. Its transition to the $\left(\bar{1}\bar{1}2\bar{2}\right)$ plane (see Figure 5d) was noted. Striations parallel to the $\left(\bar{1}\bar{1}2\bar{3}\right)$ plane were found (see Figure 5e). However, as the lateral expansion of area (3) continued, the formation of the non-polar $\left(1\bar{2}10\right)$ plane was not observed.

An extremely important fact is that the formation of semi-polar planes reduces the lateral growth rate. This was evidenced by the shape of the interfaces between areas (3) and (4). While the interface between areas (2) and (4) was only slightly inclined to the $\left(000\bar{1}\right)$ plane, the angle of inclination of the interface between areas (3) and (4) increased. This was due to a change in the relationship between the growth rates of individual areas. The change of the interface angle to the $\left(000\bar{1}\right)$ plane was due to the change in growth rate after the formation of both the $\left(\bar{1}01\bar{2}\right)$ and $\left(\bar{1}\bar{1}2\bar{4}\right)$ planes (see Figure 5e). The growth rate on these planes was lower in comparison to the expansion of area (2) and comparable to or slightly higher than the growth rate in the $\left[000\bar{1}\right]$ crystallographic direction. It also seemed that in the case of the lateral expansion of the crystal toward the $\left[10\bar{1}0\;\right]$ direction, the transition between the semi-polar planes was related to the gradual reduction in the growth rate. This was observed in terms of the increasing angle of inclination to the $\left(000\bar{1}\right)$ plane of the interface between areas (3) and (4). Finally, the transition ended with the formation of the stable $\left(\bar{1}010\right)$ plane, where the growth rate was the lowest, as shown by earlier studies [15].

Figure 11 presents a schematic model of the growth along the $\left[000\bar{1}\right]$ and $\left[\bar{1}010\right]$ directions on a lens seed. The sequence of the process was analyzed for the $\left(11\bar{2}0\right)$ plane slice. The shape of the crystal in the initial phase of growth is presented in Figure 11a. At this stage, under steady growth conditions, the crystallization began at interface A (dark dashed line) and was carried out perpendicularly to it (area (2)). The average growth rate perpendicular to the curvature of the seed (GR1) was higher than the average growth rate in the $\left[000\bar{1}\right]$ direction. A higher growth rate (GR1, see Figure 11) led to the formation of two new crystallographic planes. Above the center of the seed, where the off-cut is close to 0°, restoration of the stable $\left(000\bar{1}\right)$ plane occurred. Near the edge of the seed, where the off-cut of the seed was higher than 25°, the $\left(\bar{1}01\bar{2}\right)$ plane started to develop. In the next phase, the crystallization run was carried out on three surfaces: on the front parallel to the curvature of the seed (with GR1), the recovered $\left(000\bar{1}\right)$ plane (with GR2) and the semi-polar $\left(\bar{1}01\bar{2}\right)$ plane (with GR3). At this stage, a slower growth rate (GR3) was noted in the $\left[\bar{1}01\bar{2}\right]$ crystallographic direction. The slowest growth rate (GR2) was found to be in the $\left[000\bar{1}\right]$ direction. Reducing the growth rate on these planes caused their surface area to expand during the process. This was accompanied by the disappearance of the surface parallel to the curvature of the seed. As the crystallization process continued, the expansion of the $\left(000\bar{1}\right)$ surface was observed (see Figure 11a). Note that the restoration of the $\left(000\bar{1}\right)$ plane began in the center of the crystal and, over time, took place in regions far away from the center. The change in the direction of crystallization, and thus the growth rate (in area (4)), resulted in varying concentrations of unintentionally incorporated impurities. Both facts explained the formation of interface B (see Figure 11b) inclined toward the $\left(000\bar{1}\right)$ plane. This happened until the growth on the surface parallel to the curvature of the seed disappeared completely. Further growth was only realized on the $\left(\bar{1}01\bar{2}\right)$ and $\left(000\bar{1}\right)$ planes, as shown in Figure 11b. Figure 11c shows the further lateral expansion of area (3) and vertical expansion of area (4). Near the edge of the seed, in the lower part of area (3), a transition between the growth on two semi-polar $\left(\bar{1}01\bar{2}\right)$ and $\left(\bar{1}01\bar{1}\right)$ planes was observed. The formation of the $\left(\bar{1}010\right)$ plane also appeared. Since that moment, the lateral expansion of the crystal was realized on two semi-polar $\left(\bar{1}01\bar{2}\right)$ and $\left(\bar{1}01\bar{1}\right)$ planes, as well as one non-polar $\left(\bar{1}010\right)$ plane. Considering only lateral crystallization, it can be stated that the growth rate in the $\left[\bar{1}01\bar{2}\right]$ direction (GR3) was higher than in the $\left[\bar{1}01\bar{1}\right]$ direction (GR4). The lowest growth rate (GR5) was observed on the stable non-polar $\left(\bar{1}010\right)$ plane. It should also be noted that compared with the $\left[000\bar{1}\right]$ direction, the growth rate toward the $\left[\bar{1}01\bar{2}\right]$ direction was higher, and toward $\left[\bar{1}01\bar{1}\right]$, it was lower. Therefore, interface C in the initial phase, when growth in area (3) was realized on the $\left(\bar{1}01\bar{2}\right)$ plane, was inclined to the $\left(000\bar{1}\right)$ plane at an angle of approximately 1 ° Differences in growth rates caused the fastest-growing planes to disappear. Thus, the expansion of the $\left(\bar{1}01\bar{1}\right)$ plane replaced the growth on the $\left(\bar{1}01\bar{2}\right)$ plane. As a consequence, interface C began to curve upward, which is clearly presented in Figure 11d. Finally, lateral expansion was carried out on two slowly growing planes: $\left(\bar{1}01\bar{1}\right)$ and $\left(\bar{1}010\right)$.

Figure 12 shows the sequence of the growth process from the perspective of the $\left(\bar{1}100\right)$ plane. The crystal’s shape in the initial phase of the growth run is presented in Figure 12a. The crystallization started where interface A was marked and was carried out perpendicularly to this interface. A part of the crystal grown with the high rate GR1 led to the formation of two crystallographic planes. Above the center of the seed, where the off-cut was close to 0°, restoration of the stable $\left(000\bar{1}\right)$ plane occurred. Near the edge of the seed, where the off-cut of the seed started to exceed 25°, the $\left(\bar{1}\bar{1}2\bar{4}\right)$ plane was developed. Next, crystallization was carried out on three surfaces: parallel to the curvature of the seed (with the highest GR1), on the semi-polar $\left(\bar{1}\bar{1}2\bar{4}\right)$ plane (with lower GR2) and on the recovered $\left(000\bar{1}\right)$ plane (the lowest GR3).

In time, an expansion of the $\left(000\bar{1}\right)$ plane was observed (see Figure 12b). It should be noted that the recovery of the $\left(000\bar{1}\right)$ plane began in the center of the crystal and then expanded toward its edges. In addition, the change in the direction of crystallization, and thus, the growth rate resulted in the change in the concentration of unintentionally incorporated impurities. Both facts explain the formation of interface B inclined toward the $\left(000\bar{1}\right)$ plane (see Figure 12c).

Expansion of the crystal in the lateral direction was realized by growth on the newly formed $\left(\bar{1}\bar{1}2\bar{4}\right)$ plane. The formation of this plane was observed close to the edge of the seed. In area (3), the transition from the $\left(\bar{1}\bar{1}2\bar{4}\right)$ plane to the $\left(\bar{1}\bar{1}2\bar{2}\right)$ plane with a higher angle of inclination to the $\left(000\bar{1}\right)$ plane was observed. This fact was confirmed by the striations visible in area (3) shown in Figure 5d. In addition, the observed change in the crystallization plane was accompanied by a change in the growth rate. The growth rate on the $\left(\bar{1}\bar{1}2\bar{2}\right)$ semi-polar plane (GR4) was lower than the growth rate on the $\left(\bar{1}\bar{1}2\bar{4}\right)$ plane (GR2). The expansion of the crystal beyond the edge of the seed was accompanied by the formation of the $\left(0001\right)$ plane.

The formation and expansion of the $\left(000\bar{1}\right)$ plane was carried out until the surface (parallel to the curvature of the seed) disappeared (Figure 12c). Then, the growth was realized only perpendicularly to the stable $\left(000\bar{1}\right)$ plane, as well as on the newly developed semi-polar planes: $\left(\bar{1}\bar{1}2\bar{4}\right)$ and $\left(\bar{1}\bar{1}2\bar{2}\right)$, located in the upper and lower parts of area (3), respectively. However, it should be noted that the growth rate on the developed semi-polar planes was higher than the growth rate on the $\left(000\bar{1}\right)$ plane. As a consequence, changes in the shape of interface C between areas (3) and (4) were noted (see Figure 12d). The slope of interface C started to increase relative to the $\left(000\bar{1}\right)$ plane but it did not bend upward, as was visible in the analyzed $\left(1\bar{2}10\right)$ slice (see Figure 3b). Herein, for growth analysis of the $\left(\bar{1}100\right)$ slice, no matter which of the developed crystallographic semi-polar planes were formed, the growth rates on them were always similar or slightly higher than in the $\left[000\bar{1}\right]$ direction.

The presented model can help to predict the crystal’s shape if the growth process had lasted longer. The expected result is presented in Figure 13. Interface C would have a different curvature for both analyzed slices. For the $\left(1\bar{2}10\right)$ slice until the semi-polar $\left(\bar{1}01\bar{1}\right)$ plane is formed, interface C will bend upward (see Figure 13a). After the formation of the $\left(\bar{1}01\bar{1}\right)$ semi-polar plane, the interface slope will change. Then, due to the fact that GR2 > GR4, an inclination of interface C to the -*c* plane will be observed. For the $\left(\bar{1}100\right)$ slice, interface C will be constantly inclined to the $\left(000\bar{1}\right)$ plane (see Figure 13b).

The deterioration of the structural quality in areas (2), (3) and (4) near the edge of the crystal (AP IV), as seen in the HRXRD measurements of both slices, was a consequence of the growth anisotropy. Measurements of the lattice parameters indicated a strong lattice mismatch, represented as relative strain, between areas (3) and (4). The mismatch can be defined by the following equation:Δε_(3-4)_ = (*a*_(3)_ − *a*_(4)_)/(*a*_(4)_)(1)
where *a*_(3)_ and *a*_(4)_ are the values of the *a*-lattice parameters measured in areas (3) and (4), respectively. The calculated Δε_(3-4)_ was equal to 2.0 × 10^−4^ and 3.8 × 10^−4^ for the $\left(\bar{1}100\right)$ and $\left(1\bar{2}10\right)$ slices, respectively. These values were relatively high and may have had a strong impact on the structural quality of the crystal. Moreover, such a mismatch could lead to structural deterioration in both areas at the edge grown in lateral and vertical directions. The difference in the lattice parameters measured in area (3) and (4) was caused by varying concentrations of unintentionally incorporated impurities. It is especially visible in the case of O, whose concentration was higher in area (3) than in area (4), as verified using SIMS. This also affected the free carrier concentration and, as a result, led to faster photo-etching of area (3). The strong influence of the lattice parameter mismatch on the strain in the crystal was described by Lucznik et al. [28] and Amilusik et al. [29] for the homoepitaxial growth of HVPE-GaN. Thus, the presence of strong strain between the mentioned areas was most probably the cause of the appearance of cracks in layer (4), as shown in Figure 3a,b. However, no cracks were observed in the crystal’s volume after the growth. They appeared during the preparation of the slices as a result of strain relaxation.

In addition to the findings of the SIMS measurements, the increase for the *a*-lattice parameters and the larger photo-etching rate for areas (3), other features of increased impurity incorporation were observed for these crystal regions. On one hand, it is the bright green luminescence under UV illumination. On the other hand, a distinctive diffuse scattering intensity was observed in the HRXRD experiments. The cause of diffuse scattering near Bragg reflections could have been lattice distortions in crystalline materials due to point defects and/or precipitates [30]. For the analysis, log–log plots of the diffuse scattering intensity measured around the $1\bar{2}10$ reflection and the $\bar{3}300$ reflection of the *a*-plane slice and the *m*-plane slice, respectively, were prepared (Figure A5 and Figure A6 in Appendix D). For both reflections, the intensity decreased according to I~*q*^−2^ (where I is the X-ray intensity and *q* is the deviation of the scattering vector *H* from the reciprocal lattice vector *G*) [31]. This diffuse scattering, called Huang diffuse scattering, is due to the far field of elastic distortions around defects and occurs at relatively small wave vectors. In the $1\bar{2}10$ reflection, a small range at higher wave vectors additionally indicated another decrease according to I~*q*^−4^ and originated in the vicinity of defects where the distortions were strong. This kind of diffuse scattering is named Stokes–Wilson scattering [32]. Huang and Stokes-Wilson diffuse scattering are clear indicators of point defects and their clusters. This clearly confirmed the strongly increased incorporation of impurities, as was also detected by other methods, e.g., SIMS. Moreover, based on these HRXRD observations, the bright diffuse contrasts for the laterally grown GaN crystal regions (3) in the XRT images (Figure 6, Figure 7, Figure A1 and Figure A2) could be clearly explained by the diffuse scattering.

As mentioned earlier, area (2) could be divided into two sub-areas (see Figure 3). The creation of sub-area (2b) was treated as a disturbance of growth in area (2). SIMS measurements showed that the O concentration was almost an order of magnitude higher in sub-area (2b) than in sub-area (2a). The lower incorporation of O in (2a) might have been the result of fast growth carried out perpendicularly to the curvature of the seed. As a consequence, a low carrier concentration led to fast etching during the PE. The HRXRD results showed a lower structural quality of area (2). However, it should be noted that this measurement was made without taking into account the exact position on the crystal. Only the SIMS measurements were performed in such a manner to investigate sub-areas (2a) and (2b). Significant differences in O concentrations between these sub-areas might have induced strong stress and deterioration of structural quality that were visible in the higher FWHM and FWTM values.

In the case of growth on a lenticular seed, a variable off-cut increasing toward the edge of the seed was present. Assuming a constant supersaturation on the surface of the growing crystal, the growth rates are determined by the surface kinetics. Increasing the off-cut angle determines the change in the growth mode on the surface. A low off-cut angle (higher step length), which is present close to the seed center, promotes the growth in the step propagation growth mode. A higher off-cut (lower length of the steps—shorter diffusion length), which is present in the location further away from the center of the seed, can promote a shift from the bilayer step propagation to a step-bunching growth mode [33]. The formation of high steps may have led to the formation of semi-polar planes. Their creation may have been the beginning of the formation of sub-area (2b), which disturbed the growth that was realized perpendicularly to the curvature of the seed (area 2a) (see Figure 4a and Figure 5a).

Closer to the edge of the crystal, where the off-cut of the seed was higher, the time needed for the restoration of the $\left(000\bar{1}\right)$ plane was longer. A higher growth rate and also higher off-cut favored the destabilization of the crystallization front and the possibility of inclusions appearing, as well as changes in the crystallization direction and growth mode. The formation of sub-area 2b may have resulted in the annihilation of dislocations and lowering the EPD (see Figure 10b,d). The lowest EPD was found on the surface close to the edge of the crystal (see Figure 10c,d) located exactly above area (3). The XRT analysis (see Figure 6 and Figure 7) showed the first refraction of the dislocation at the interface between areas (2a) and (2b). The second refraction was observed at the interface between areas (2b) and (3). Because the growth in area (3) was on successive semi-polar planes, with a greater angle of inclination to the $\left(000\bar{1}\right)$ plane, propagation of the dislocations to this plane was not possible. Thus, the crystal grown above area (3) was characterized by the lowest EPD.

The model of vertical and lateral expansion of the crystal (Figure 11 and Figure 12), together with the XRT analysis (Figure 6 and Figure 7), allowed for explaining the observed lateral gradient of the EPD presented in Figure 10. The higher EPD in the central part of the crystal (see Figure 10a) resulted from the fact that the crystallization front in the initial phase of the growth process was slightly inclined to the $\left(000\bar{1}\right)$ plane. In area (4), all dislocations propagated almost perpendicular to the $\left(000\bar{1}\right)$ plane. In area (2a), the propagation occurred perpendicularly to the curvature of the lens seed (see Figure 6). Moreover, in area (2), due to the relatively high growth rate, some disturbances might have appeared at the crystallization front. Their consequence was a change in the growth mode and the formation of area (2b). The change in the growth mode may have caused a variation in the propagation of dislocations and, finally, their annihilation. Some dislocations originating from the seed were refracted at the interface between areas (1) and (2a). Another refraction of the dislocations occurred at the interface between regions (2) and (4). No growth disturbance was observed in the central part of the crystal, where area (2a) was small. A reconstruction of the $\left(000\bar{1}\right)$ plane was observed very early. Finally, no significant shift or separation in the dislocations was observed in the central part of the crystal after the second refraction (see Appendix A).

## 5. Conclusions

In this paper, an analysis of crystallization of GaN on a native seed of a lenticular shape, with a varying off-cut of the surface, is presented. It was established that under standard basic ammonothermal growth conditions, the growing crystal tried to restore its equilibrium shape via the creation of low-energy planes. In the early stages of growth, this was accomplished by a very fast growth perpendicular to the curvature of the seed. For an off-cut close to 0°, the $\left(000\bar{1}\right)$ plane formation occurred quickly. For off-cuts close to 25° and higher, the formation of semi-polar planes, different for the two distinguished crystallographic directions, was observed. Initially, lateral growth was carried out on the developed semi-polar planes with the smallest inclination angle to the $\left(000\bar{1}\right)$ plane. Further growth was accompanied by a transformation of these semi-polar planes into other planes with a higher angle of inclination. Eventually, the lateral growth and the transformation between the semi-polar planes ended with the formation of the non-polar $\left\{\bar{1}010\right\}$ and polar $\left\{0001\right\}$ planes. The formation of six planes from the $\left\{\bar{1}010\right\}$ crystallographic family dramatically slowed down the lateral growth of the crystal. The growth rate on the developed semi-polar planes was different for the $\langle \bar{1}\bar{1}20\rangle$ and $\langle \bar{1}010\rangle$ set of directions. In comparison to the $\left[000\bar{1}\right]$ direction, growth in the $\langle \bar{1}010\rangle$ directions was generally slower, but for the $\langle \bar{1}\bar{1}20\rangle$ family of directions, it was equal or slightly faster. This fact determined the final shape of the obtained crystal.

Reaching the equilibrium shape also determined the structural quality of the grown crystal. The lateral growth realized on semi-polar planes forced the refraction of threading dislocations. Thus, the material growing in the vertical direction directly above this laterally growth part was characterized by a lower density of defects compared with the material growing directly above the seed.

The measured differences in the concentrations of impurities between the areas of lateral and vertical growth were the cause of varying lattice parameters and the reason for the formation of stress at the interface between these areas. The appearance of stress caused a deterioration in the structural quality in particular areas and, in extreme situations, may result in the appearance of cracks.

With this work, we were able to gain in-depth knowledge about the growth of GaN using the basic ammonothermal growth technique, which will contribute to the continuous improvement of the material quality for GaN-based devices.

## Figures and Tables

**Figure 1 materials-15-04621-f001:**
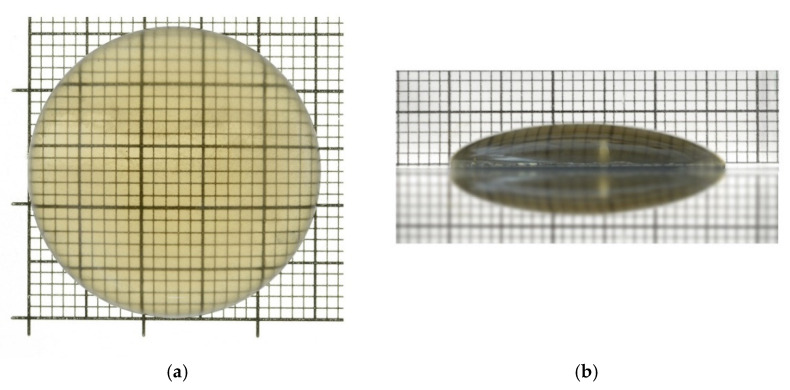
Selected and specially prepared Am-GaN seed with a lenticular shape: (**a**) top view of the $\left(000\bar{1}\right)$ plane; (**b**) side view. The thickness of the seed at its center was 4.1 mm; radii of the surface curvature varied between 21 and 25 mm; grid line 1 mm.

**Figure 2 materials-15-04621-f002:**
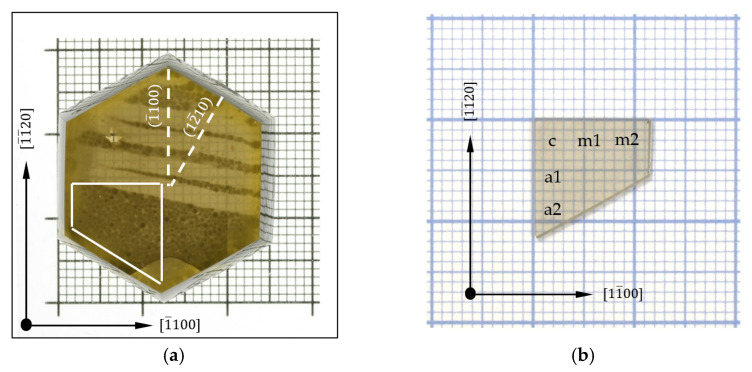
(**a**) Top view of a hexagonal Am-GaN crystal grown on a lens seed; white dashed lines indicate the location of slicing for samples with non-polar planes; white solid trapezoid represents a sample (see **b**) obtained by cutting a slice in the $\left(000\bar{1}\right)$ plane. (**b**) Slice of the sample that was cut and prepared for DSE; view on the $\left(0001\right)$ plane; EPD was determined at indicated points; grid line 1 mm.

**Figure 3 materials-15-04621-f003:**
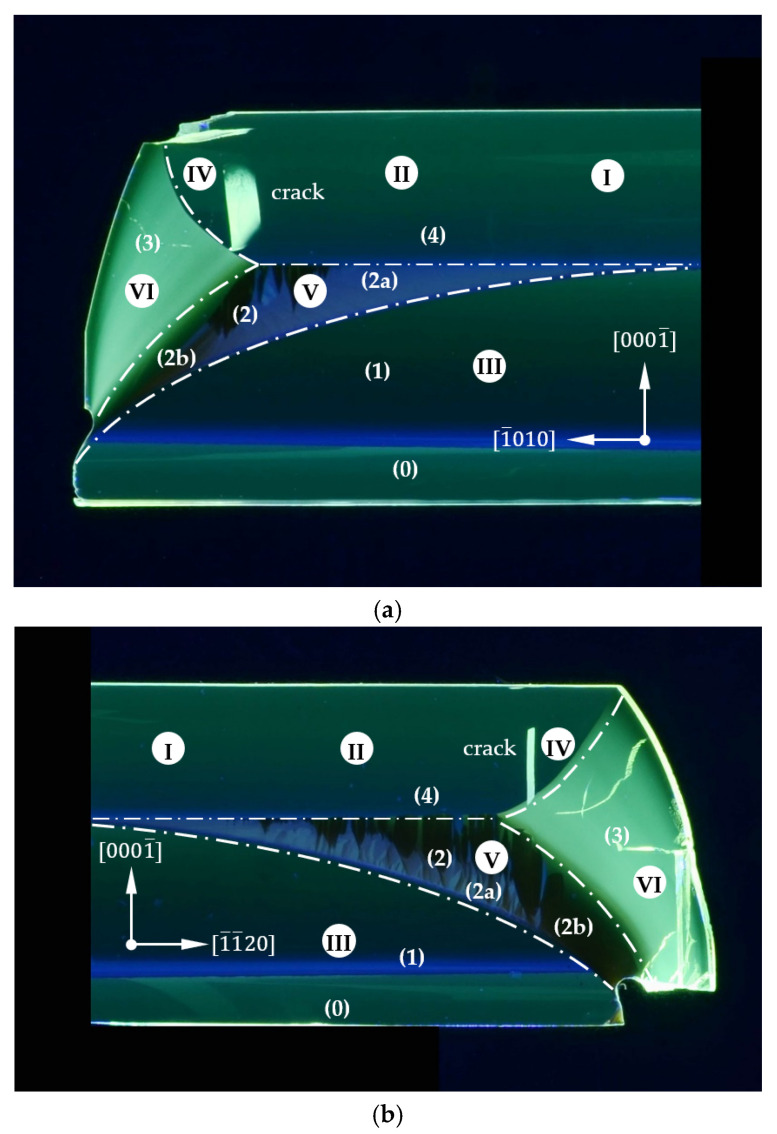
Slices of an Am-GaN crystal grown on the lenticular seed. OM images gathered under UV illumination: (**a**) $\left(1\bar{2}10\right)$ plane and (**b**)$\;\left(\bar{1}100\right)$ plane. Five areas separated by clear interfaces were named with Arabic numerals in brackets: (0)—pre-seed; (1)—lens seed; (2)—area of the $\left(000\bar{1}\right)$ plane recovery; two sub-areas: (2a) and (2b), differing in the intensity of UV luminescence, were distinguished; (3)—area of lateral growth; (4) area of GaN grown in the $\left[000\bar{1}\right]$ direction. One crack was clearly visible. Roman numerals (I–VI) were used to represent the six locations of HRXRD and SIMS measurements called “analysis points (APs)”.

**Figure 4 materials-15-04621-f004:**
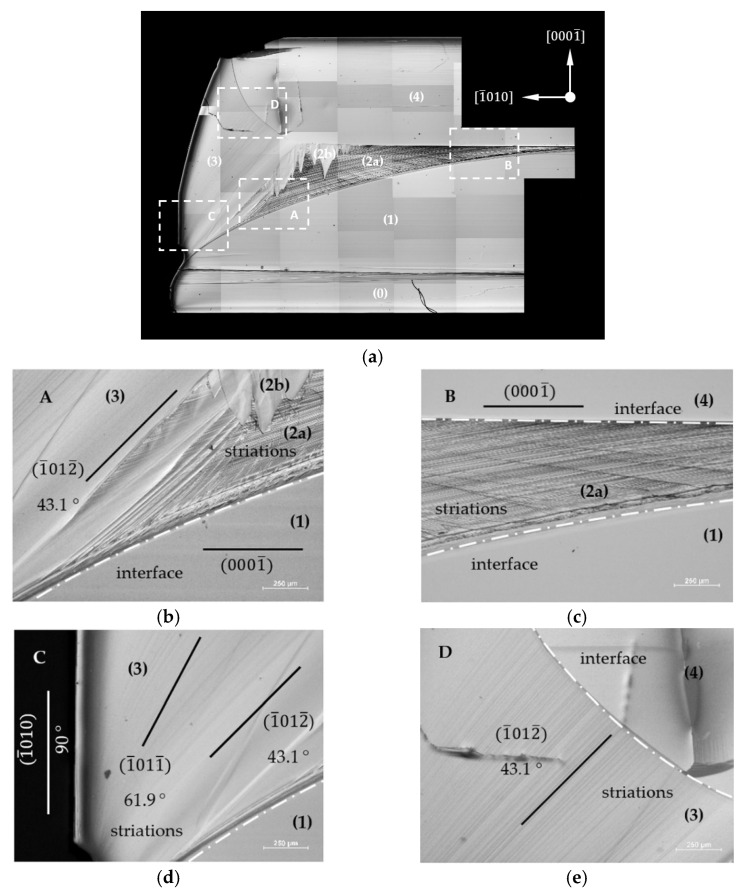
Images of the $\left(1\bar{2}10\right)$ slice of an Am-GaN crystal after the PE: (**a**) View of the entire sample; white rectangles (A–D) indicate the sectors chosen for further detailed analysis. (**b**) Sector A; (**c**) sector B; (**d**) sector C; (**e**) sector D. Crystallographic planes parallel to the visible striations and their angle of inclination to the $\left(000\bar{1}\right)$ plane are marked: the $\left(\bar{1}01\bar{2}\right)$ plane and 43.1°, the $\left(\bar{1}01\bar{1}\right)$ plane and 61.9°, and the $\left(\bar{1}010\right)$ plane and 90°; interfaces between different areas of the crystal (see **a**) are marked with white dashed lines.

**Figure 5 materials-15-04621-f005:**
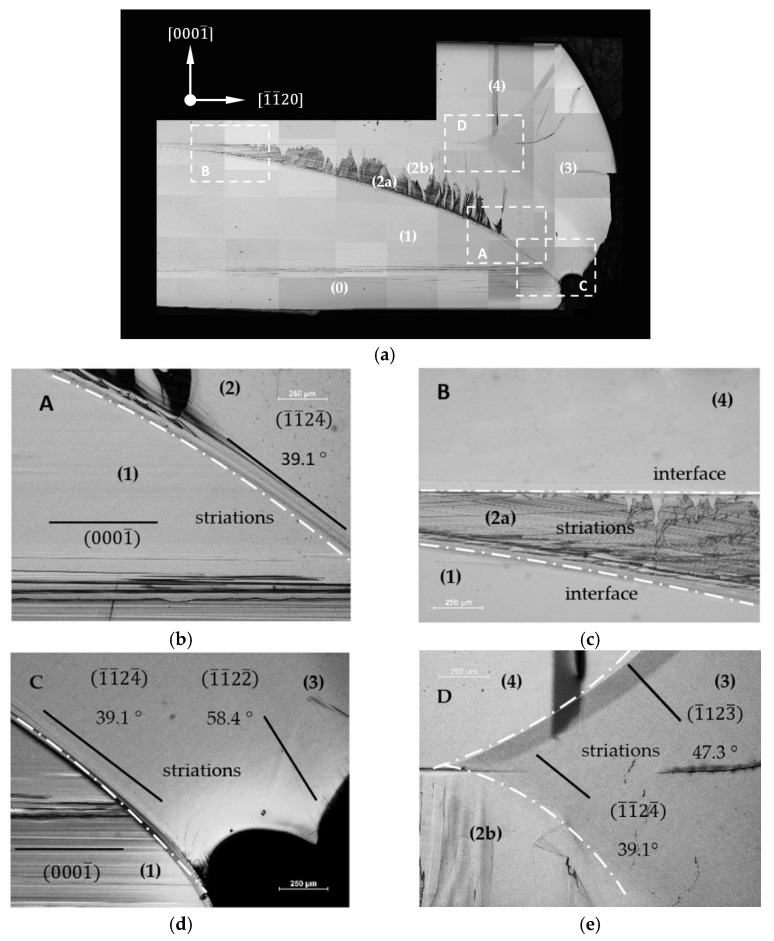
Images of the $\left(\bar{1}\bar{1}00\right)$ slice of an Am-GaN crystal after the PE: (**a**) View of the entire sample; white rectangles (A–D) indicate sectors for further detailed analysis. (**b**) Sector A; (**c**) sector B; (**d**) sector C; (**e**) sector D. Crystallographic planes parallel to the visible striations and their angle of inclination to the $\left(000\bar{1}\right)$ plane are marked: the $\left(\bar{1}\bar{1}2\bar{4}\right)$ plane and 39.1°, the $\left(\bar{1}\bar{1}2\bar{2}\right)$ plane and 58.4°, and the $\left(\bar{1}\bar{1}2\bar{3}\right)$ plane and 47.3°; the interfaces between different areas of the crystal (see Figure 3a) are marked with white dashed lines.

**Figure 6 materials-15-04621-f006:**
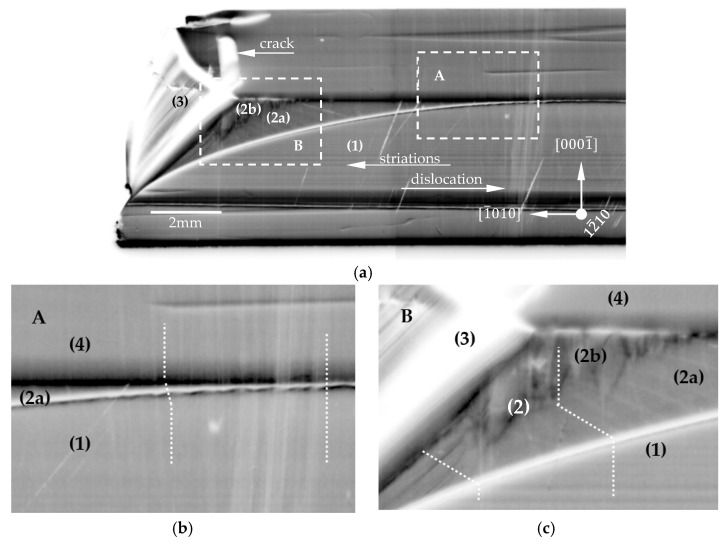
X-ray topographs of the $\left(1\bar{2}10\right)$ slice. For the imaging, the $000\bar{2}$ reflection (Cu-Kα_1_ radiation) was used. (**a**) Overview topograph of the $\left(1\bar{2}10\right)$ plane; five areas separated by clear interfaces were visible, as seen in Figure 3a,b; one crack was also present. Magnified images of sectors presented in Figure 6a: (**b**) sector A, where dislocations from area (1) went almost straight to area (4), and (**c**) sector B, with two-stage dislocation refraction in area (2) (considered as the sum of areas 2a and 2b) and one stage refraction from area (1) to (3).

**Figure 7 materials-15-04621-f007:**
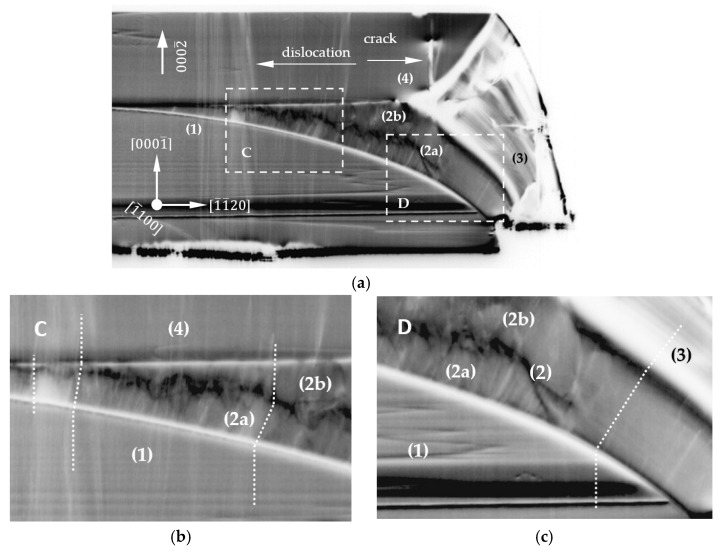
X-ray topographs of the $\left(\bar{1}100\right)$ slice. To image the $000\bar{2}$ reflection, Cu-Kα_1_ radiation was used: (**a**) overview topograph of the $\left(\bar{1}100\right)$ plane; five areas separated by clear interfaces were visible, as in Figure 3a,b; one crack was also present. Magnified images of sectors presented in Figure 7a: (**b**) sector C, where the dislocations from area (1) went straight to area (4) and the two-stage dislocation refraction in area (2) (considered as the sum of areas 2a and 2b); (**c**) sector D, where the two-stage dislocation refraction went from area (1) to area (3).

**Figure 8 materials-15-04621-f008:**
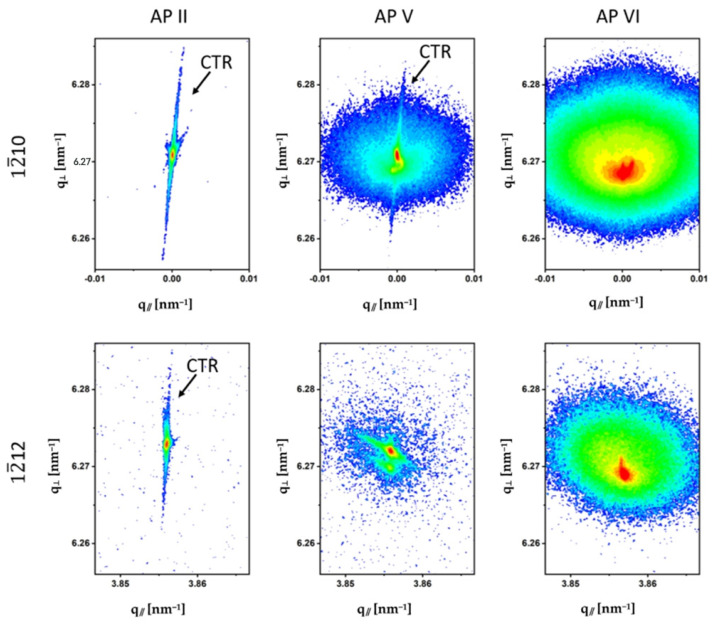
RSMs of the symmetrical $1\bar{2}10$ reflection and the asymmetrical $1\bar{2}12$ reflection performed on the $\left(1\bar{2}10\right)$ slice. Positions of the crystal truncation rods (CTRs) are indicated on the maps.

**Figure 9 materials-15-04621-f009:**
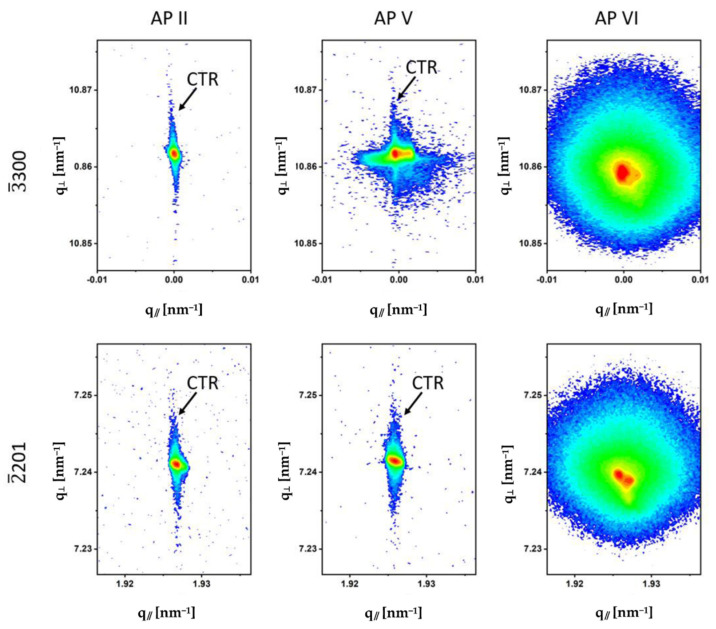
RSMs of the symmetrical $\bar{3}300$ reflection and the asymmetrical $\bar{2}201$ reflection performed on the $\left(\bar{1}100\right)$ slice. Positions of the crystal truncation rods (CTR) are indicated on the maps.

**Figure 10 materials-15-04621-f010:**
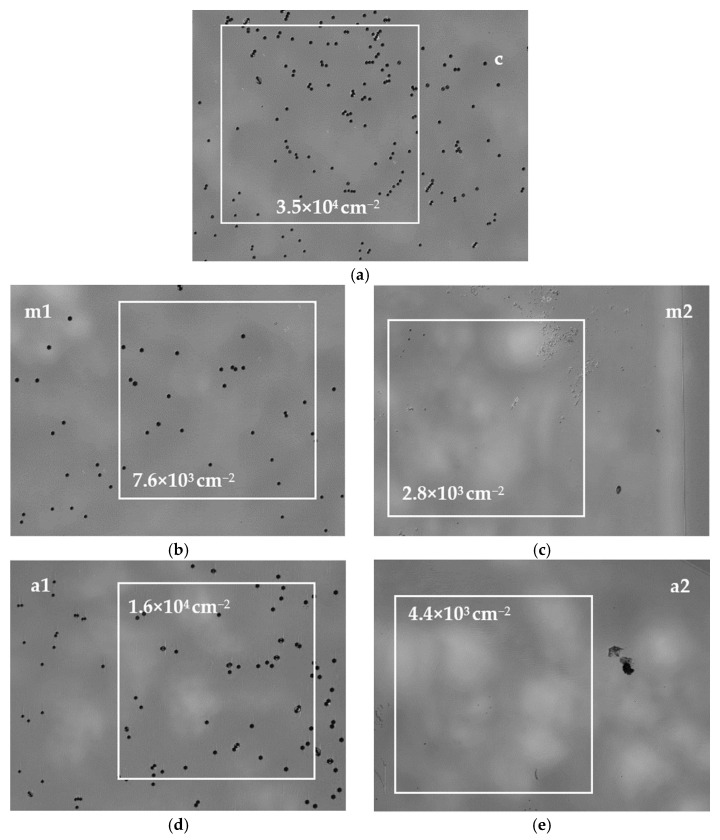
DIC optical images of the $\left(0001\right)$ plane after the DSE; the EPD at five points indicated in Figure 2b: (**a**) c, (**b**) m1, (**c**) m2, (**d**) a1 and (**e**) a2; white squares indicate areas of the EPD estimation.

**Figure 11 materials-15-04621-f011:**
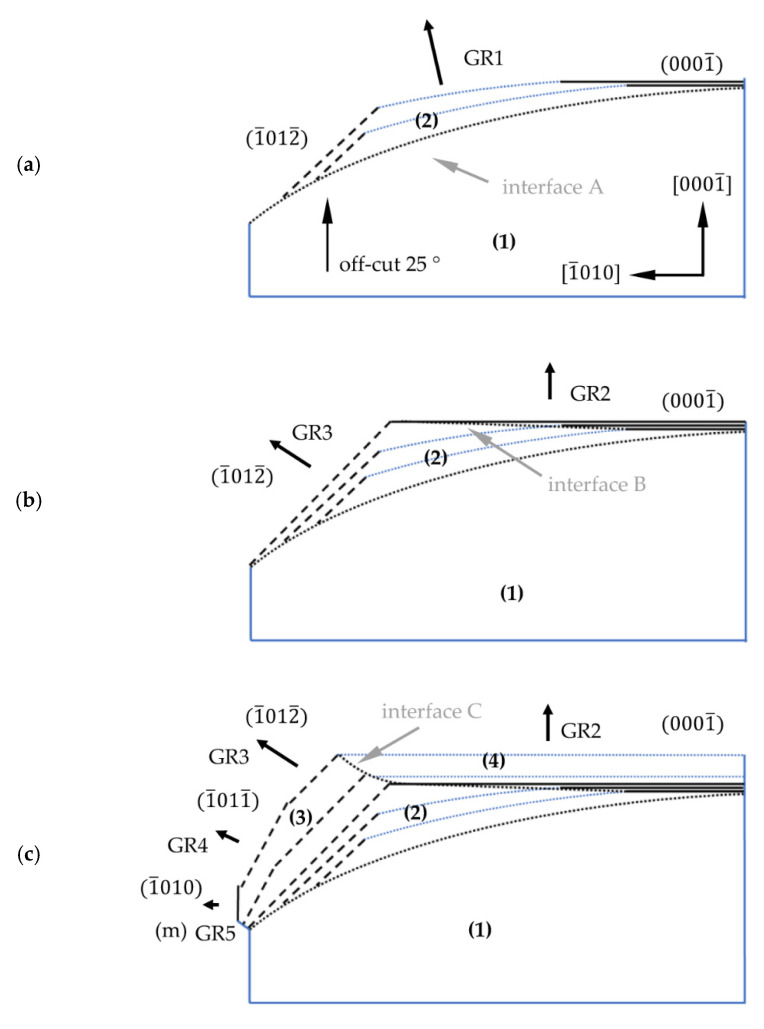
Model of Am-GaN growth on a lenticular seed along the $\left[000\bar{1}\right]$ and $\left[\bar{1}010\right]$ directions with a view of the $\left(1\bar{2}10\right)$ plane: (**a**) the first stage of the growth—recovery of the $\left(000\bar{1}\right)$ plane and formation of the $\left(\bar{1}01\bar{2}\right)$ semi-polar plane; (**b**) growth on the recovered $\left(000\bar{1}\right)$ plane and growth on the $\left(\bar{1}01\bar{2}\right)$ plane; (**c**) growth on the $\left(000\bar{1}\right)$, $\left(\bar{1}01\bar{2}\right)$, $\left(\bar{1}01\bar{1}\right)$ and $\left(\bar{1}010\right)$ planes; (**d**) expansion of the $\left(\bar{1}010\right)$ plane and transition from growth of the $\left(\bar{1}01\bar{2}\right)$ to the $\left(\bar{1}01\bar{1}\right)$ plane.

**Figure 12 materials-15-04621-f012:**
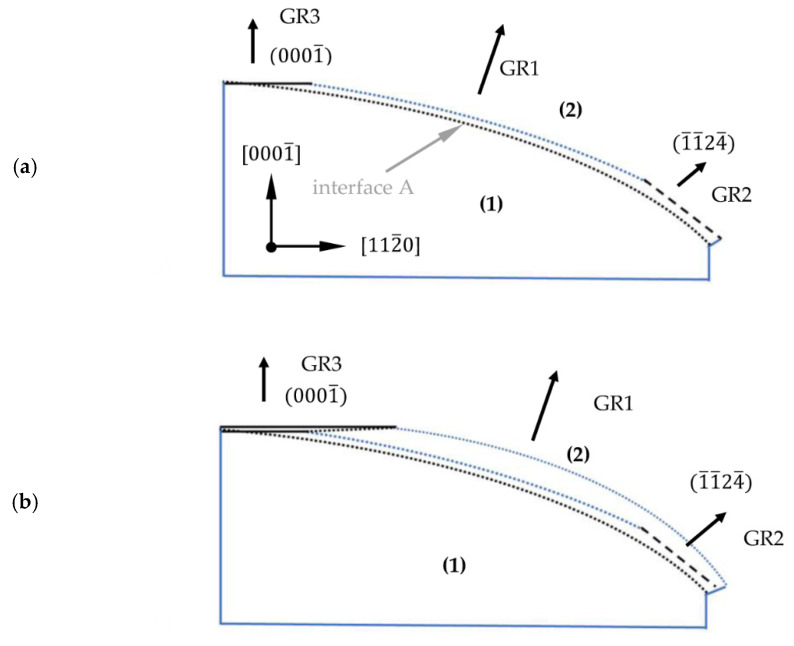
Model of Am-GaN growth on a lenticular seed along the $\left[000\bar{1}\right]$ and $\left[\bar{1}\bar{1}20\right]$ directions with a view of the $\left(\bar{1}100\right)$ plane: (**a**) the first stage of the growth—recovery of the $\left(000\bar{1}\right)$ plane and formation of the semi-polar $\left(\bar{1}\bar{1}2\bar{4}\right)$ plane; (**b**) growth on the recovered $\left(000\bar{1}\right)$ plane and growth on the $\left(\bar{1}\bar{1}2\bar{4}\right)$ plane; (**c**) growth on the $\left(000\bar{1}\right)$ and $\left(\bar{1}\bar{1}2\bar{4}\right)$ planes with a transition between the $\left(\bar{1}\bar{1}2\bar{4}\right)$ and $\left(\bar{1}\bar{1}2\bar{2}\right)$ semi-polar planes and the formation of the $\left(0001\right)$ plane; (**d**) growth on the $\left(000\bar{1}\right)$ and $\left(\bar{1}\bar{1}2\bar{2}\right)$ planes with the transition between the $\left(\bar{1}\bar{1}2\bar{4}\right)$ and $\left(\bar{1}\bar{1}2\bar{3}\right)$ semi-polar planes and the expansion of the $\left(0001\right)$ plane.

**Figure 13 materials-15-04621-f013:**
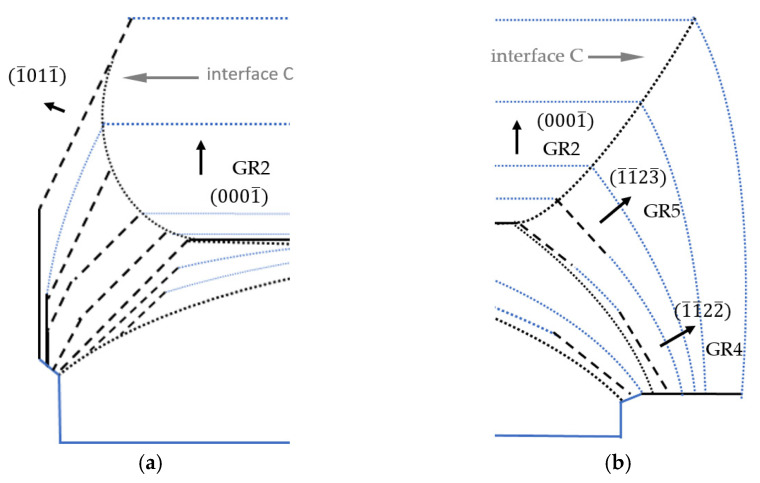
Predicted behavior of interface C for a longer growth process and the resulting shape of the crystal: (**a**) view on the $\left(1\bar{2}10\right)$ plane and (**b**) view on the $\left(\bar{1}100\right)$ plane.

**Table 1 materials-15-04621-t001:** Concentrations of the main impurities (donors and acceptors) measured using SIMS on the $\left(1\bar{2}10\right)$ slice at six analysis points (APs) (indicated as Roman numerals in circles in Figure 3a) and the corresponding growth areas (indicated as Arabian numerals in brackets in Figure 3a).

$\left(1\bar{2}10\right)\;\mathrm{AP}/\mathrm{Area}$	[H] (cm^−3^)	[O] (cm^−3^)	[Si] (cm^−3^)	[Na] (cm^−3^)	[Mg] (cm^−3^)	[Fe] (cm^−3^)	[Mn] (cm^−3^)	[Zn] (cm^−3^)
III/(1)	1.6 × 10^19^	9.7 × 10^18^	1.1 × 10^17^	6.1 × 10^16^	1.0 × 10^17^	1.9 × 10^15^	1.0 × 10^17^	2.8 × 10^17^
V/(2a)	2.6 × 10^18^	2.3 × 10^18^	1.9 × 10^17^	8.3 × 10^16^	6.3 × 10^15^	2.4 × 10^14^	8.0 × 10^15^	9.1 × 10^16^
V/(2b)	2.9 × 10^19^	8.9 × 10^18^	2.6 × 10^17^	4.5 × 10^17^	2.0 × 10^16^	3.9 × 10^14^	2.3 × 10^16^	1.3 × 10^17^
I/(4)	1.3 × 10^19^	4.4 × 10^18^	1.0 × 10^17^	1.8 × 10^16^	2.8 × 10^16^	2.5 × 10^14^	2.7 × 10^16^	1.0 × 10^17^
IV/(4)	1.1 × 10^19^	3.8 × 10^18^	8.7 × 10^17^	1.7 × 10^16^	2.4 × 10^16^	6.3 × 10^14^	2.2 × 10^16^	1.0 × 10^17^
VI/(3)	1.2 × 10^20^	6.1 × 10^18^	4.8 × 10^17^	1.1 × 10^18^	6.4 × 10^16^	7.2 × 10^15^	1.3 × 10^17^	1.3 × 10^17^

**Table 2 materials-15-04621-t002:** Concentrations of the main impurities (donors and acceptors) measured using SIMS on the $\left(\bar{1}100\right)$ slice at six analysis points (APs) (indicated as Roman numerals in circles in Figure 3b) and the corresponding growth areas (indicated as Arabian numerals in brackets in Figure 3b).

$\left(\bar{1}100\right)\;\mathrm{AP}/\mathrm{Area}$	[H] (cm^−3^)	[O] (cm^−3^)	[Si] (cm^−3^)	[Na] (cm^−3^)	[Mg] (cm^−3^)	[Fe] (cm^−3^)	[Mn] (cm^−3^)	[Zn] (cm^−3^)
III/(1)	1.2 × 10^19^	9.6 × 10^18^	9.2 × 10^16^	4.0 × 10^16^	5.4 × 10^16^	1.9 × 10^15^	7.2 × 10^16^	2.4 × 10^17^
V/(2a)	2.6 × 10^18^	1.3 × 10^18^	2.7 × 10^17^	3.1 × 10^16^	4.1 × 10^15^	4.9 × 10^14^	7.6 × 10^15^	1.2 × 10^17^
V/(2b)	2.1 × 10^19^	1.0 × 10^19^	1.5 × 10^17^	8.0 × 10^16^	2.8 × 10^16^	8.8 × 10^14^	3.1 × 10^16^	1.1 × 10^17^
I/(4)	6.4 × 10^18^	4.2 × 10^18^	6.9 × 10^16^	1.6 × 10^16^	2.1 × 10^16^	5.9 × 10^14^	1.9 × 10^16^	9.8 × 10^16^
IV/(4)	4.3 × 10^18^	3.1 × 10^18^	5.1 × 10^16^	1.2 × 10^16^	1.5 × 10^16^	7.5 × 10^14^	1.3 × 10^16^	9.8 × 10^16^
VI/(3)	1.3 × 10^20^	9.8 × 10^18^	5.6 × 10^17^	1.9 × 10^18^	7.9 × 10^16^	8.1 × 10^15^	1.5 × 10^17^	2.2 × 10^17^

## Data Availability

Not applicable.

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
