# Peer review of "Fundamental Studies on Crystallization and Reaching the Equilibrium Shape in Basic Ammonothermal Method: Growth on a Native Lenticular Seed"

_materials, 2022, doi:10.3390/ma15134621_

Round 1

Author Response

Dear Reviewer,

We would like to thank you for all the comments and corrections which really helped us to improve the manuscript. All of them have been considered and applied. They also allowed us to find and eliminate several other minor mistakes. We would like to thank you for acceptance of the manuscript for publication.

Reviewer 2 Report

In this paper, the authors give a detailed investigation of the basic ammonothermal growth process on a native seed with a lenticular shape by means of optical microscopy, x-ray topography and high-resolution x-ray diffraction, secondary ion mass spectrometry. Based on the analysis of the above characterization data, the schematic models were built to clearly show the evolution process of crystal growth. The paper reads well and these findings are very helpful to understand the mechanism of crystal growth by ammonothermal method. There is a question to discuss. On Lines 632 and 633, the authors said low carrier concentration leads to fast etching during PE. In general, the high carrier concentration helps to accelerate the etching. I don’t know the etching mechanism of PE. Could the authors explain it?

Author Response

The answer can be found in the attached file.

Reviewer 3 Report

This is a well written and important contribution.  No changes are required.

The multiple planes for growth clearly can explain the dislocation morphology which is often observed.

Author Response

Dear Reviewer,

We would like to thank you for all comments and for acceptance of the manuscript for publication. All language errors have been corrected.